# Perceived Barriers and Benefits of COVID-19 Testing among Pacific Islanders on Guam

**DOI:** 10.3390/ijerph20136302

**Published:** 2023-07-05

**Authors:** Rachael T. Leon Guerrero, Angelina G. Mummert, Dareon C. Rios, Niza C. Mian, Teofila P. Cruz, Chathura Siriwardhana, Richard Yanagihara

**Affiliations:** 1Office of Research & Sponsored Programs, University of Guam, Mangilao, GU 96923, USA; 2Department of Tropical Medicine, Medical Microbiology and Pharmacology, John A. Burns School of Medicine, University of Hawaii at Manoa, Honolulu, HI 96813, USA

**Keywords:** Guam, Pacific Islanders, CHamorus, Micronesians, COVID-19

## Abstract

Pacific Islanders residing in the U.S. Affiliated Pacific Islands have had among the highest COVID-19-associated morbidity and mortality rates in the U.S. To reduce this disparity, we conducted a study to increase the reach and uptake of COVID-19 testing in Guam. Participants, who completed a pre-survey on demographics, health status, history of COVID-19 testing and vaccination, access to COVID-19 testing, sources of COVID-19 information, and knowledge and attitudes towards COVID-19 test results and transmission, were invited to attend an online educational session about COVID-19 testing and transmission and to complete a post-survey. There were significant positive changes between pre- and post-survey in knowledge and perceptions about COVID-19 testing and transmission, but changes were not necessarily due to exposure to the educational session. Compared to CHamoru participants (n = 380), Other Micronesians (n = 90) were significantly less knowledgeable about COVID-19 transmission and testing, were significantly more likely to not want to know if they had COVID-19, were more likely to believe if they did have COVID-19 there was not much that could be done for them, and that they would have difficulty in getting the needed healthcare. This study is another example of disparities in health knowledge and perceptions of certain Pacific Islander groups.

## 1. Introduction

Long before the COVID-19 pandemic, the indigenous people of the three territories of the U.S. Affiliated Pacific Islands (USAPI), namely Guam, Commonwealth of the Northern Mariana Islands, and American Samoa, already suffered disproportionately from the dual burden of communicable and non-communicable diseases [1,2]. Furthermore, Pacific Islander communities have poor access to health care, lack health insurance [3], face racial discrimination, lack trust in the health care system [4], and suffer from generations-long historical trauma, implicit bias, poverty, and colonialism [5]. In addition, Pacific Islanders are underrepresented in biomedical research and reports, and most of the available research aggregates data from Pacific Islanders and Asian Americans, making it nearly impossible to distinguish the actual disparities between and within Pacific Islander and Asian communities [6].

Part of Micronesia, Guam is the southernmost island of the Mariana Archipelago in the northwestern Pacific Ocean, lying approximately 3700 miles west of Hawaii, 6000 miles west of California, and 1300 miles southeast of Japan. CHamorus, the original inhabitants of Guam [7,8], are typically grouped with other Pacific Islanders in national surveys. The current population of Guam is characterized by substantial ethnic variation [9]: 37% CHamoru; 26% Filipino; 12% other Pacific Islander; 7% White; 7% other Asian; and 11% other ethnicities. This ethnic diversity evolved through centuries of colonization and migration that continues today [10]. Spanish colonization of Guam between 1668 and 1898, followed by the subsequent U.S. possession, has led to substantial ethnic and cultural admixture.

Pacific Islanders residing in the USAPI, including people of the sovereign nations that have a Compact of Free Association treaty with the U.S. (Federated States of Micronesia, Republic of the Marshall Islands, and Republic of Palau), have had among the highest COVID-19-associated morbidity and mortality rates in the U.S. This group also suffers from medical co-morbidities known to increase their risk of severe COVID-19. In addition, Pacific Islanders tend to live in multi-generational or multi-family overcrowded housing and have low-paying service jobs that expose them to the infected public. Between March 2020 and December 2022, Guam experienced dramatic surges in COVID-19, with triple-digit daily counts and more than 10,000 cases. Other Micronesian groups, such as Chuukese, Palauans, Pohnpeians, Marshallese, and Yapese, who comprise only 10.8% of Guam’s population, accounted for 8% of COVID-19 cases and 19.5% of COVID-19 deaths [11].

To help reduce disparities in COVID-19 incidence and mortality among Pacific Islanders, the Puipuia le Ola project was conducted. Puipuia le Ola, which means “protecting life” in Samoan, and Prutehi I Linala in the CHamoru language, was a dual-site collaborative project between the University of Hawaii at Manoa and the University of Guam, and was funded in the fall of 2020 by the National Institutes of Health (NIH) Rapid Acceleration of Diagnostics among Underserved Populations (RADx-UP) program (grant #3P30GM114737-05S1). The funded project aimed to: (1) promote COVID-19 testing among Pacific Islanders (particularly Samoans, CHamorus, and other Micronesians in Hawaii and Guam) using culturally relevant community-tailored strategies with culturally and linguistically appropriate educational materials; and (2) identify the factors associated with COVID-19 testing, including trusted sources of COVID-19 information. This project was the first to develop a program to increase COVID-19 testing among these underserved groups. The purpose of this paper is to present the study results from Guam.

## 2. Materials and Methods

### 2.1. Participant Eligibility

Individuals were eligible to participate in the study if they met all of the following criteria: (1) age 18 years or older; (2) self-identified as Pacific Islander; (3) lived on Guam for at least three months; (4) had never tested positive for COVID-19; (5) did not have a bleeding disorder, immune deficiency, or autoimmune disease; and (6) able to give written informed consent.

Prospective participants were screened in-person during outreach events or screened remotely via phone, email, or Google Forms to determine eligibility. Contact information for all eligible individuals was collected and a Unique ID code was assigned—by the research staff—to de-identify participants once informed consent forms were signed and received.

### 2.2. Participant Recruitment

The study was conducted from 7 June 2021 to 14 July 2022. Respondent Driven Sampling (RDS) [12] was used for participant recruitment. RDS is a recruitment and link-tracing sampling method that starts with an initial sample of study participants who serve as “seeds” [12]. Seeds recruit acquaintances who comprise the sample’s “first wave”. The first wave recruits the second wave until the desired sample size is reached. One of the main advantages of RDS is the ability to rapidly recruit community participants through the existing social networks of participants [12]. RDS has several unique features to minimize bias in simple chain-referral or snowball sampling [12]. For this study, the recruitment target was 800 participants in Guam.

Initially, a few participants served as RDS seeds and were instructed to recruit others utilizing the study’s recruitment coupons, as previously described [13]. Recruitment coupons included details such as contact information for the study, expiration date, and a Unique ID code, which was used to track the network size of respondents [13]. The initial participant was categorized as a seed, and their recruits were then organized into waves 1, 2, and 3, respectively. The maximum number of recruits by a recruiter was limited to three.

### 2.3. Pre-Survey

Eligible participants who provided written informed consent were administered a pre-survey, which included questions about demographics (sex, race/ethnicity, language, employment, education level, and income), health status, history of COVID-19 testing, history of COVID-19 vaccination, access to COVID-19 testing, sources of COVID-19 information, and knowledge and attitudes towards COVID-19 test results and transmission. All survey questions came from the common data element (CDE) set for the NIH RADx-UP project [14]. Study participants who self-identified as Chuukese, Kosraean, Palauan, Pohnpeian, or Yapese were categorized as “Other Micronesian” for data analysis purposes since these groups were small and to distinguish them from the indigenous CHamoru of Guam.

### 2.4. Educational Session

Study participants who completed the pre-survey were invited to attend an optional 45 min online educational session held via Zoom. Participants received reminder phone calls along with an email link for their scheduled Zoom session. Cameras were disabled, and names were de-identified during the educational sessions. Presentations were modified and tailored towards Pacific Islanders on Guam based on current COVID-19 statistics. The purpose of the educational session was to increase knowledge about COVID-19 and to provide detailed information about the importance of COVID-19 testing. Interactive questions about COVID-19 were asked, and participants were encouraged to respond by typing their answers into the Zoom chat box. At the end of each session, participants were offered free COVID-19 testing.

### 2.5. COVID-19 Testing

Free COVID-19 testing was offered throughout the study. Diagnostic testing initially consisted of COVID-19 reverse transcription-polymerase chain reaction (RT-PCR) test. All eligible participants signed a specimen collection consent form before being tested. All sample specimens were sent to Hawaii’s Diagnostic Laboratory Services to be processed. Test results were received by a contracted medical provider who informed participants about their results. RT-PCR tests were discontinued on 12 March 2022, and the Ellume COVID-19 Home Test kits (Ellume, Frederick, MD, USA) were made available to interested participants. Research staff tracked the distribution of test kits and followed up on test results. The Guam Department of Public Health Social Services (GDPHSS) was notified of all positive tests during the study.

### 2.6. Post-Survey

Up to three months after the completion of the pre-survey and/or educational session, participants were asked to complete a post-survey either electronically or in hard copy. The post-survey aimed to assess each participant’s stage of change toward COVID-19 testing uptake and to evaluate the impact (if any) of the study’s COVID-19 educational session. The post-survey consisted of selected questions from the pre-survey.

### 2.7. Participant Enrollment

A total of 505 participants were enrolled into the study and completed the pre-survey. Of these, 248 participants (49.1%) completed the post-survey and 103 (20.4%) completed the COVID-19 educational session. Table 1 indicates how many participants were recruited under each wave. Followed by the general RDS strategy, it was expected that with each successive wave, there would be an exponential growth in the number of participants recruited. However, in this study only 22% of participants were recruited in wave 1 with lower recruitment rates following subsequent waves. Since RDS was not an efficient strategy to recruit participants, we resorted to other community outreach strategies to inform Pacific Islanders about the study, generate interest, and recruit participants. These included the use of social media, community WhatsApp chat groups, radio talk show interviews, press releases, local newspaper articles, emails and Zoom and phone calls to community-based and faith-based organizations, village mayors, and other Pacific Islander community organizations, and participation at local community events for in-person recruitment, such as working in tandem with the COVID-19 vaccination clinic sites at local shopping centers and public health clinics. Approximately 72% of participants were through direct recruitment by the study team, which means the sample was dominated by seeds.

Ethical approval for this project was granted by the University of Guam Committee on Human Research Subjects (CHRS#20-172); written informed consent was obtained from all participants, in accordance with the Declaration of Helsinki.

### 2.8. Incentives

Study participants received a gift certificate of $20 to a local grocery store for each completed survey, and an additional $20 gift certificate for attending the COVID-19 educational session. Participants were strongly encouraged to recruit others within their social networks and received an additional $5 per recruit. Thus, each participant could receive up to $75 in gift-card incentives for participating in the study.

### 2.9. Data Entry

Participants used their Unique ID code to complete the pre- and post-surveys (either online or on hard copy). Participants who opted to complete the surveys online were given a link by the research staff and entered their survey data electronically into a REDCap database system. For participants who completed their surveys via hard-copy, research staff entered their (hard-copy) survey data manually into the REDCap database system.

### 2.10. Statistical Analysis

Participant demographics and basic characteristics were analyzed using descriptive statistics. Categorical data were summarized using frequencies and percentages, while continuous data were presented as means and standard deviations. To account for the sampling method employed in this study, an RDS scheme, RDS-weighted estimators were utilized to summarize the survey results related to knowledge and attitude on COVID-19 and COVID-19 testing. The RDS-II weighted estimates were accompanied by 95% confidence intervals (95% CI) and standard errors (SE) [15,16].

The outcome measures of knowledge and attitude on COVID-19 testing were treated as ordinal categorical variables with response options ranging from a low to a high level (e.g., Strongly Disagree, Disagree, Neither Disagree nor Agree, Agree, Strongly Agree). To analyze changes in these ordinal response variables between the pre-survey and the post-survey, cumulative link mixed effect models were with a logit link function were employed. This statistical model is specifically designed to analyze ordinal response variables with multiple ordered categories. It extends the traditional cumulative link model by incorporating random effects to account for the heterogeneity or clustered structure of the data. In our analysis, random effects were included for both the participant and the RDS recruiter to address the clustering effect. The odds of observing higher-order response at the post-survey, compared to the pre-survey, were calculated and tested using the Wald test. Furthermore, an interaction effect between COVID-19 educational session attendance and survey status was examined to assess whether the changes from pre-to-post were associated with the COVID-19 education session. A sensitivity analysis was conducted by excluding data from waves 2 and 3, which had a smaller number of participants.

To investigate the associations between race/ethnicity and the observed COVID-19 knowledge and attitude variables during the pre-survey, cumulative link mixed effect models were employed, incorporating random effects for RDS recruiters.

All statistical analyses were performed using the R statistical programming software version 4.0.2. A *p*-value of <0.05 was considered to be statistically significant.

## 3. Results

### 3.1. Participant Demographics

The descriptive characteristics of study participants are summarized in Table 2. The average age of participants was 34.8 years (sem 0.587), and the majority were female (65%), 18–27 years of age (36.8%), of CHamoru ethnicity (75.3%), currently employed (63.8%), had a family with children or multi-generational family (64.7%), and had an annual income level greater than $50,000 (34.5%). In addition, participants tended to have a high level of education with the majority either having a college degree (35.1%) or at least some college, technical or vocational education (34.3%). Since the beginning of the COVID-19 pandemic in March 2020, 40.6% of participants reported that someone in their household (or themselves) had experienced a loss of employment income.

Overall, two-thirds of participants described their health status as ‘good’ or better, but 28.6% of participants described their health as fair or poor. Among participants, 17.2% had hypertension, 9.5% had diabetes, 9.1% had asthma, 2.2% had cardiovascular disease, and 10.9% had depression.

### 3.2. COVID-19 Testing

At baseline, 73.7% of participants had previously been tested for COVID-19, and of those tested, 97.9% had been tested via the nasal swab method. Of those who did not get tested for COVID-19, the most common reason was because they had not felt sick at all (65.5%) or not felt sick enough to get tested (12.7%). A small number of participants did not get tested because they believed their faith in God would protect them from COVID-19 (3.5%) or because they did not think it was safe to go to a testing location (6.3%). Overall, the majority of participants agreed/strongly agreed that it was easy to get tested for COVID-19 (67.1%), they knew where to get COVID-19 testing in their community (78.4%), and they planned to get tested as often as needed (57.9%).

Only about half of the participants were confident/very confident that negative test results meant that they did not have COVID-19 (52.2%); and a little more than half were confident/very confident that positive test results meant that they had COVID-19 (58.0%). As many as 14.0% of participants believed that if they recieved a negative COVID-19 test result, they did not have to worry about getting COVID-19; and this was especially true for Other Micronesian participants as they were 3.87 times more likely than CHamorus to believe that they did not have to worry about getting COVID-19 if they recieved a negative COVID-19 test result (OR = 3.87; CI: 1.71–6.25; *p* < 0.001). When participants were asked what they would do if they ever received a positive COVID-19 test result, most stated that they would isolate themselves (95.5%), about half would need to take off from work (49.4%), and some participants believed that they would need to be admitted to the hospital (19.4%).

The most common factors contributing to participants getting tested for COVID-19 were wanting to get treated early if they were positive (71.8%) and wanting to know that they were safe to not give COVID-19 to others (69.7%). The main factor that discouraged participants from getting tested for COVID-19 was that if they tested positive, public health officials would need to contact all of the people who they had been in contact with (43.1%). This was especially true for Other Micronesian participants as they were more likely to report this as a barrier for COVID-19 testing (OR = 2.66; CI: 1.61–4.39; *p* < 0.0001). Other Micronesian participants were less likely to want to know if they had COVID-19 (OR = 4.16; CI: 2.52–6.87; *p* < 0.0001), which is most likely related to the aversion of having public health officials notify their contacts if positive for COVID-19.

Other Micronesians believed that if they did not have COVID-19 symptoms, that they did not need to be tested (OR = 1.85; CI: 1.15–2.98; *p* < 0.01), demonstrating less awareness about COVID-19 than other ethnic groups. Other Micronesians were more likely to believe that if they did have COVID-19, there was not much that could be done for them (OR = 2.56; CI: 1.60–4.11; *p* < 0.0001), and they were more likely to believe that it would be difficult to get the healthcare that they would need if they had COVID-19 (OR = 3.64; CI: 2.23–5.94; *p* < 0.0001).

Results of the post-test indicated a significant change in knowledge and perceptions about COVID-19 testing. Participants were significantly less likely to believe that if they tested positive, they would need to be admitted to the hospital (OR = 0.30; CI: 0.16–0.56; *p* < 0.0001). In addition, participants were significantly more likely to agree that there were benefits to get tested for COVID-19, such as reducing the worry that they might have COVID-19, knowing if they are safe not to give COVID-19 to others, and getting treated early if, they were positive. Participants were significantly less likely to believe that they would experience discomfort from being tested (OR = 0.78; CI: 0.76–0.79; *p* < 0.0001). Most importantly, participants were significantly less likely to not want to know if they had COVID-19 (OR = 0.24; CI: 0.12–0.48; *p* < 0.0001).

### 3.3. Knowledge of COVID-19 Transmission

At baseline, most participants were aware that COVID-19 could be transmitted by coming in close contact with an infected person who had symptoms (86.8%), an infected person not showing symptoms (86.7%), or having contact with surfaces that an infected person had touched (82.1%). However, Other Micronesian participants were significantly less aware of COVID-19 transmission. Compared to CHamoru participants, the Other Micronesian participants were significantly less likely to know that a person could get infected with COVID-19 if they have close contact with an infected person who had symptoms (OR = 0.11; CI: 0.05–0.23; *p* < 0.0001); had come in close contact with an infected person who did not show symptoms (OR = 0.16; CI: 0.08–0.33; *p* < 0.0001); and had come in contact with surfaces that an infected person had touched (OR= 0.38; CI: 0.21–0.70; *p* < 0.002).

Post-survey results indicated a significant improvement in knowledge of COVID-19 transmission. Participants were more likely to know that COVID-19 could be transmitted by coming in close contact with an infected person with symptoms (OR = 6.83; CI: 2.09–22.43; *p* < 0.0015).

### 3.4. COVID-19 Vaccination

At baseline, nearly all of the participants (92.7%) had received at least one dose of the COVID-19 vaccine. Of those who had not yet received the COVID-19 vaccine, 14.2% indicated that they would likely get a COVID-19 vaccine in the future. Participants were also asked about reasons why they would or would not like to get the COVID-19 vaccine. The biggest concern about getting the COVID-19 vaccine was possible side effects (34.7%) and not knowing enough about how well the vaccine worked (21.3%). The main reason why participants wanted to get the vaccine was to keep themselves (74.9%) and their family (86.9%) safe.

### 3.5. Information Sources and Trusted Sources of COVID-19 Information

Of the 505 participants who completed the pre-survey, 69.6% responded that they had heard a “great deal” about COVID-19, 26.1% heard “some”, and 4.3% heard “not much”. Slightly more (74.9%) felt they had sufficient information about COVID-19. However, a quarter (25.1%) of those surveyed did not think they had sufficient information about COVID-19.

Participants reported receiving COVID-19 information from various local and national sources. The top five local sources included: GDPHSS/Public Health website (77.3%), family and friends (62.1%), posted information in the community (59.8%), doctor/health care providers (45.6%), and coworkers/employers (36.7%). Social media platforms were another avenue employed by participants seeking information about COVID-19. Of the six listed social media sources, Facebook (54.0%) was most frequently selected, followed by Instagram (41.1%), YouTube (34.5%), Twitter (12.0%), and TikTok (9.5%).

The most trusted sources of COVID-19 information were GDPHSS and its website (65.1%), doctors/health care providers (62.1%), Guam’s local Task Force [JIC/Governor’s COVID-19 Pandemic Task Force] established to provide COVID-19 pandemic guidance to government executive leaders (58.4%), the U.S. Center for Disease Control website (59.9%), and the U.S. Coronavirus Task Force (47.9%)

### 3.6. Impact of Educational Session

A total of 103 participants attended the educational session, of which 79.6% were CHamoru and 20.3% were Other Micronesian. For those participants who attended the educational session, their post-survey responses were compared to their pre-survey responses. Overall, 82% of participants agreed/strongly agreed that the educational session was a positive experience. In addition, participants (77.6%) expressed interest in enrolling in future research projects with the University of Guam.

There were no significant differences in the perceived benefits of testing and knowledge of COVID-19 between those who attended the educational session versus those who did not. Although the knowledge of COVID-19 improved in the post-survey for study participants in general. This change was not necessarily due to the educational session, as there was no statistically significant difference when the results were compared to those who did not participate in the educational session. However, there was an improvement in the perceptions of the benefits of testing between pre-survey and post-survey; 69.6% of participants agreed/strongly agreed that this project successfully increased COVID-19 testing for their friends and family. More than half of the participants (57.7%) received a COVID-19 test after they enrolled in the study. Furthermore, 97% of participants said they were tested via the nasal swab method also used during the project’s drive-through testing. Additionally, participants (94.8%) expressed that as a result of participating in this study, they would practice COVID-19 transmission prevention by wearing a mask, washing hands frequently, and keeping a 6-feet social distance.

## 4. Discussion

This is one of few studies that looks at the knowledge, perceived barriers, and perceived benefits of COVID-19 testing and transmission among Pacific Islanders, in particular CHamorus and Other Micronesians, living on Guam. In this study, there were significant positive changes, pre-survey versus post-survey, in the knowledge and perceptions about COVID-19 testing and transmission, although the changes were not necessarily due to exposure to the educational session. These changes may be due to external factors, such as losing a loved one, contracting COVID-19, and COVID-19 knowledge saturation. At baseline, most participants felt they had already heard a great deal about COVID-19. However, one-fourth of participants did not think they had sufficient information about COVID-19, and thus most likely benefited from the information that they gleaned by participating in the study.

At baseline, Other Micronesians demonstrated a significantly lower level of knowledge about COVID-19 testing and transmission compared to CHamorus, the largest ethnic group in Guam and in this study. This disparity in paramount health knowledge is not new as others in Guam have documented that health information is not reaching underserved and underrepresented groups, such as Chuukese, Palauans, Pohnpeians, Kosraean, and Yapese [17,18]. In addition, the Other Micronesians group reported that they did not even want to know if they had COVID-19 because they were significantly more likely to believe that there was not much that could be done for them if they were sick, that it would be difficult to get the healthcare that they would need if they had COVID-19, and that they did not want the local health department contacting all of the people who they had been in contact with. Other Micronesians in this study may have been hesitant to have their COVID-19 status disclosed to family and friends via contact tracing for fear of being ostracized by their community or experiencing discrimination. In addition, Other Micronesians may lack trust in the healthcare system due to historical and systemic issues of mistreatment and lack of access to healthcare. Hattori-Uchima [19] reported that Other Micronesians on Guam, Chuukese in particular, had difficulty obtaining the healthcare they needed due to financial concerns, communication issues, and mistrust; they tend to not seek preventive healthcare services. Several studies have shown that Micronesian migrants often identify feelings of mistrust related to negative experience with healthcare workers [19], and feelings of mistreatment and discrimination when seeking help from health, education, and public welfare agencies [20].

For the Other Micronesian participants in this study, the lower level of knowledge about COVID-19 testing and transmission, in addition to their attitude of not even wanting to get tested for COVID-19 because they did not want to know the results likely explains why the number of COVID-19 deaths during the pandemic were so high in the Other Micronesian group, who comprise only 10.8% of Guam’s population, but accounted for 19.5% of COVID-19 deaths on Guam [11]. Micronesians already face significant health disparities including high rates of non-communicable diseases, such as diabetes and heart disease [1,2,5], and the current study confirms the continued health disparities in this underserved community in Guam.

In order to reach community members of Other Micronesians and Other Pacific Islanders on Guam, a partnership was formed with the Mañe’lu Micronesian Resource Center One-Stop Shop (MRCOSS) and Neechuumeres Chuukese Women of Guam organization. MRCOSS is a local nonprofit organization who provides informational and educational services in the home language of people moving to Guam from the Freely Associated States. Neechuumeres is a community-based grassroots organization aimed at supporting the social, cultural, spiritual, and economic well-being of Chuukese women in their communities. Recruitment and follow-up for these ethnic groups were mainly conducted in-person alongside the local outreach events. Various members of MRCOSS and Neechuumeres assisted in English translation to the field researchers during recruitment. The majority of Chuukese, Pohnpeian, Palauan, Marshallese, Yapese, and Kosraean participants were recruited through these local partnerships.

About one-quarter of study participants described their general health status as “fair” or “poor”. This is higher than what was seen in a survey by National Center for Health Statistics (NCHS) [21,22] where 15.5% of Native Hawaiians and Pacific Islanders reported their health stats as “fair or poor”, which was higher than the U.S. average of 12%. In this NCHS survey, Native Hawaiians and Pacific Islander adults were more likely than U.S. adults from all other ethnic groups to be in “fair or poor” health [21,22]. The fact that participants in this study perceived their health to be even worse than other Pacific Islanders and other ethnic groups in the U.S. may be due to the timing of the current study during the COVID-19 pandemic, but it does indicate a possible disparity in perceived and real health status among the participants in this study.

In general, study participants reported that they relied on local and national health department websites for up-to-date information about COVID-19. This may have been due to COVID-19 information that was disseminated by health and government officials at daily press briefings led by the Governor of Guam and members of the COVID-19 Pandemic Task Force during the period of mandated restrictions on in-person interactions during the pandemic. These daily briefings were televised and live-streamed on social media platforms, and the GDPHSS website included daily updated COVID-19 data and critical health information. The Joint Information Center (JIC) of the Guam Homeland Security worked closely with local agencies such as GDPHSS to coordinate the response to the pandemic and keep the community well informed [23]. Local communications included daily JIC reports posted on their website, while the GDPHSS website featured a COVID dashboard. Many of the village mayors in Guam posted these reports on their individual village community centers and on social media (WhatsApp) platforms. WhatsApp became the primary means of how village residents communicated with village officials, and each other on COVID-19-related alerts and information. The unprecedented, imposed isolation prompted people to seek information through online sources and social media like never before. This further builds on a previous study on health information seeking in Guam, which revealed that the internet was the first source of information about health and medical topics for Guam residents [17] and that geographical isolation Guam and relatively limited medical resources most likely accounted for the greater reliance on online information.

In this study, the most trusted local COVID-19 information source was the GDPHSS and its website (65%) and the most trusted national government source was the CDC (60%). Participants in the present study had a much higher level of trust in the national and local health departments compared to a recent national survey, which showed that only one-third of U.S. adults [24] highly trusted the CDC for information about COVID-19 and only one-quarter of U.S. adults trusted state/local health departments. SteelFisher and co-workers reported that public trust in the CDC was related primarily to beliefs in their scientific expertise, whereas trust in local public health agencies was more related to their provision of direct, compassionate care [24]. Adults on Guam may have more trust in their local public health department as GDPHSS has a long history of working closely with the local community to address health issues and providing culturally relevant health information and resources, contributing to trust.

The COVID-19 vaccination rate was fairly high, as most of the participants (92.7%) in this study had received a COVID-19 vaccine at baseline. One of the reasons for this high vaccination rate could be due access; Guam had access to a steady supply of COVID-19 vaccines from the U.S. Government, which helped to ensure that there were enough vaccines available for everyone who wanted one. The GDPHSS launched extensive public health campaigns to encourage people to get vaccinated for COVID-19; and the strong community partnerships between the GDPHSS, healthcare providers, and community organizations helped promote COVID-19 vaccine uptake and addressed barriers to vaccination. Another possible reason may be related to Guam’s relatively high risk of COVID-19 due to its geography, military presence, and tourism industry making people on Guam more willing to get vaccinated for COVID-19 in order to protect themselves and their community from the virus [25]. Finally, Pacific Islander cultures, in general, often prioritize community and family; getting vaccinated may be seen as a way to protect and care for loved ones. Hattori-Uchima reported that one of the few health behaviors that Chuukese women sought out for themselves and their children were vaccinations [19].

Since the current study included only a small number of participants recruited in waves 2 and 3, we also conducted a sensitivity analysis on pre-to-post changes, excluding these participants. However, the results, including the magnitude of effects, directions, significances, and overall conclusion, were consistent with the initial data analysis. Please note that these results are not presented in the article.

There were several limitations in this study. First, most participants (66.5%) were between the ages of 18–40 years. This contrasts with the 2020 Census data for Guam, which reports that only 33% of the population is between the ages of 18–40 years. Thus, the study results may not be truly representative of COVID-19 transmission knowledge and attitudes of adults living on Guam. Most of the participant recruitment for this study was online due to the COVID-19 pandemic restrictions in place at that time of the study and individuals from younger age groups were more likely to respond positively to filling out the online survey, thus skewing the recruitment of participants to younger individuals.

A second limitation is that only 20% of study participants engaged in the educational session, which was conducted online via Zoom platform. It is very possible that participants did not know how to use the Zoom platform or did not feel comfortable with that mode of education and opted not to participate. Mau and colleagues [2] reported that Native Hawaiians and Pacific Islanders prefer group-based health education approaches, which may explain why few Pacific Islanders in the study participated in the online educational session.

## 5. Conclusions

Despite limitations, participants in this study did experience positive improvement in their perceived benefits and knowledge about COVID-19 testing and transmission. More importantly, we found that Other Micronesians on Guam continue to suffer from disparities in health information and knowledge compared to CHamorus in Guam, and Other Micronesians tend to believe that they cannot get the healthcare that they need if they are sick. This study of knowledge, perceived barriers, and perceived benefits of COVID-19 testing and transmission among CHamorus and Other Micronesians living on Guam, is paramount to understanding the health disparities of its people and health care delivery in crisis and for the future. More work is needed to improve health literacy and increase trust between the local health departments and Other Micronesian groups in Guam.

## Figures and Tables

**Table 1 ijerph-20-06302-t001:** RDS Recruitment Counts.

Wave	Count	Percent (%)
0 (seeds)	362	71.7
1	111	22.0
2	29	5.7
3	3	0.6

**Table 2 ijerph-20-06302-t002:** Demographic Characteristics Guam Participants (n = 505).

Characteristic	Frequency	Percent (%)
Sex		
Male	173	34.8
Female	331	65.0
Preferred not to answer	1	0.2
Age		
18–27 years	186	36.8
28–40 years	150	29.7
41–55 years	81	16.0
56+ years	48	9.5
Unknown/Prefer not to answer	40	7.9
Ethnicity of Pacific Islanders		
CHamoru	380	75.3
Other Micronesians *	90	18.0
Chuukese	38	7.5
Palauan	24	4.7
Yapese	14	2.8
Pohnpeian	11	2.2
Kosraean	3	0.6
Other Pacific Islander **	35	6.7
Current employment status		
Working now/employed	322	63.8
Unemployed/looking for work	48	9.5
Only temporarily laid off/sick leave/maternity	4	0.8
Retired	18	3.6
Student	79	15.6
Stay-at-home/keeping house	16	3.2
Disabled	2	0.4
Prefer not to answer/do not know	6	1.2
Annual Household Income (pre-pandemic in 2019)		
<$25,000	137	27.1
$25,000–$49,999	81	16.0
$50,000–$74,999	66	13.1
>$75,000	108	21.4
Prefer not to answer/do not know	113	22.4
Highest Level of Education Attainment		
12th grade or less	26	5.1
High school graduate or GED	124	24.6
Some college/technical/vocational	173	34.3
College/bachelor’s degree	108	21.4
Other advanced degree	69	13.7
Prefer not to answer/do not know	5	1.0
Family at Home		
Single/just me	59	11.7
Living with partner, no children	79	15.7
Family with children	246	48.8
Family with 3–4 generations (parents, children, grandchildren, great-grandchildren)	80	15.9
Other	40	8.0
Description of general health status		
Very good/excellent	152	30.1
Good	200	39.6
Fair	123	24.4
Poor	21	4.2
Prefer not to answer/do not know	9	1.8
Presence of medical/health conditions		
Hypertension	87	17.2
Diabetes	48	9.5
Asthma	46	9.1
Cardiovascular disease	11	2.2
Depression	55	10.9
Other mental health disorder	23	4.6

* Chuukese, Kosraean, Palauan, Pohnpeian, Yapese. ** Includes Native Hawaiian, Samoan, Marshallese, Tongan, Fijian, and Mixed-race Pacific Islander.

## Data Availability

The datasets generated and/or analyzed during the current study are not publicly available as we are still in the process of analyzing data. However, data will be available from the corresponding author after completion of data analysis on reasonable request.

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
