# Peer review of "Perceived Barriers and Benefits of COVID-19 Testing among Pacific Islanders on Guam"

_ijerph, 2023, doi:10.3390/ijerph20136302_

Round 1
Reviewer 1 Report
· The introduction mentions that the Puipuia le Ola project was developed to reduce disparities in health due to COVID among Pacific Islanders. However, the information regarding this project is presented in the M&M section. Therefore, the introduction is weak and lacks a good rationale for performing the study. Although the historic data is relevant, this information should be reduced to focus more on how COVID produced the health disparities and thus justify why the project was introduced and why the research is important to assess the effect of the project.
· Why was stated a sample size of 800 participants? Please elaborate more on this.
· “The post-survey consisted of selected questions from the pre-survey. Or (The post-survey consisted of selected questions that were identified from the pre-survey questions)” Please revise this sentence.
· I found it hard to follow the statistical analysis section. Please split into paragraphs for each test used and provide a rationale where appropriate. For instance, “cumulative link mixed effect models were applied…” and “a similar framework…”
· “A total of 505 participants were enrolled” This number falls short with respect to the target of 800 participants. Besides, the percentage of people completing the post-survey and the educational session reduced drastically. I believe that the “Participant enrollment” section should be moved to the methods to make room in the results for the remaining info. Besides, given the percentage of participants enrolled in each wave, the sampling method was a failure despite the economic incentive. Have the authors thought about removing waves 2 and 3 given the low number of participants? Also, how about performing a sensitivity analysis taking away data from these two last waves and comparing it with the results from the seeds and wave 1?
· Tables should be prepared in a similar format.
· Some parts of the results are hard to follow “This was especially true for Other Micronesian participants as they were more likely to report this as a barrier for COVID-19 testing (OR=2.66; CI: 1.61–4.39; p<0.0001). Other Micronesian participants were also far more likely to not want not to want to know if they had COVID-19 (OR=4.16; CI: 2.52–6.87; p<0.0001); were more likely to believe that if they didn’t did not have COVID-19 symptoms, that they didn’t did not need to be tested (OR=1.85; CI: 1.15–2.98; p<0.01); were more likely to believe that if they did have COVID-19, there wasn’t much that could be done for them (OR=2.56; CI: 1.60–4.11; p<0.0001); and were more likely to believe that it would be difficult to get the healthcare that they would need if they had COVID-19 (OR=3.64; CI: 2.23–5.94; p<0.0001).”
· The use of negative sentences makes it more difficult to understand the meaning “they didn’t did not have” and “didn’t did not need to be tested”
· Overall, the results section is lengthy and hard to follow given that seems more like a narrative description of the survey than an integrative summary of the findings.
· Tables are heavily populated with text. These should be synthesized.
See the comments.
Reviewer 2 Report
The study "Perceived Barriers and Benefits of COVID-19 Testing among Pacific Islanders on Guam" was reviewed. The disparities in health knowledge and perceptions of certain Pacific Islander groups were explained. There are some flaws in this study, which is to be clarified.
- Eligible participants who provided written informed consent were administered a pre-survey, which included questions about history of COVID-19 testing, history of COVID-19 vaccination, access to COVID-19 testing, sources of COVID-19 information, and knowledge and attitudes towards COVID-19 test results and transmission.
How did you prepare these questions? Was it prepared by literature review? If so, provide the reference. If you prepared the questions, did you check the reliability and validity of the study?
- Line 112 - Study participants who completed the pre-survey were invited to attend an optional 45-minute online educational session held via Zoom.
Line 134 - Up to three months after the completion of the pre-survey and/or educational session, participants were asked to complete a post-survey either electronically or in hard copy.
Conducting just 1 session of online educational session and assessing after 3 months....Why do you fix 3 months? Do you think just 1 session will have an impact on the participants? Do you have reference for the content you provide in educational session?
- Line 195 - The descriptive characteristics of study participants are summarized in Table 2. The average age was 34.8 years (sem ± 0.587).... Do you mean that SEM is 0.587?
- Lines 204-209 - How do you correlate patient's health status, co-morbidities and the the language they speak, with your study? What is the importance of these data?
- Lines 431-436 - Provide reference for these information you wrote in the discussion.
Round 2
Reviewer 1 Report
The authors put effort into preparing a revised version of their manuscript in which they properly addressed all the comments and queries to create a better well-balanced narrative. Their manuscript is ready for being accepted for publication in the journal.
Reviewer 2 Report
Dear Authors
Thank you for answering all my questions.